# The effect of shell secretion rate on Mg/Ca and Sr/Ca ratios in biogenic calcite as observed in a belemnite rostrum

Clemens Vinzenz Ullmann[1], Philip A.E. Pogge von Strandmann[2]

[1]Camborne School of Mines and Environment and Sustainability Institute, University of Exeter, Penryn, TR10 9FE, UK
[2]London Geochemistry and Isotope Centre (LOGIC), Institute of Earth and Planetary Sciences, University College London and Birkbeck, University of London, Gower Street, London, WC1E 6BT, UK

*Correspondence to*: Clemens V. Ullmann (c.v.ullmann@gmx.net)

**Abstract.** Isotopic ratios and concentrations of the alkaline earth metals Mg and Sr in biogenic calcite are of great importance as proxies for environmental parameters. In particular, the Mg/Ca ratio as a temperature proxy has had

considerable success. It is often hard to constrain, however, which parameter ultimately controls the concentration of these elements in calcite.

Here, multiple Mg/Ca and Sr/Ca transects through a belemnite rostrum of *Passaloteuthis bisulcata* (Blainville, 1827) are used to isolate the effect of calcite secretion rate on incorporation of Mg and Sr into the calcite. With increasing calcite secretion rate Mg/Ca ratios decrease and Sr/Ca ratios in the rostrum increase. In the studied specimen this effect is found to

be linear for both element ratios over a calcite secretion rate increase of ca. 150 %. Mg/Ca ratios and Sr/Ca ratios show a linear covariation with increasing relative growth rate, where a 100 % increase in growth rate leads to a (8.1 ± 0.9) % depletion in Mg and a (5.9 ± 0.7) % enrichment in Sr. The magnitude of the calcite secretion rate effect on Mg is (37 ± 4) % greater than that on Sr. These findings are qualitatively confirmed by a geochemical transect through a second rostrum of *Passaloteuthis* sp.

Growth rate effects are well-defined in rostra of *Passaloteuthis* but only account for a minor part of chemical heterogeneity. Biasing effects on palaeoenvironmental studies can be minimised by informed sampling, whereby the apex and apical line of the rostrum are avoided.

## 1 Introduction

The measurement of Mg and Sr concentrations in biogenic calcite as records of past environmental conditions and

geochemical cycles has a long tradition (e.g., Odum, 1951; Turekian, 1955, Lowenstam, 1961). Palaeotemperature estimates from element concentrations in fossil calcite in particular have been intensively studied (e.g., Pilkey and Hower, 1960; Nürnberg et al., 1996; Elderfield and Ganssen, 2000; McArthur et al., 2007).

A number of empirical studies have documented positive co-variation of Mg/Ca ratios and/or Sr/Ca ratios with temperature in biogenic calcite from belemnites (Rosales et al., 2004; McArthur et al., 2007; Li et al., 2012), bivalves (Klein et al., 1996;

Freitas et al., 2006; Wanamaker et al., 2008; Tynan et al., in press), brachiopods (Brand et al., 2013; Butler et al., 2015),

coccoliths (Stoll et al., 2001), echinoderms (Pilkey and Hower, 1960), foraminifers (Lea et al., 1999; Elderfield and Ganssen, 2000; Lear et al., 2002; de Noojer et al., 2014) and ostracods (Chivas et al., 1986; De Dekker et al., 1999). Inorganic precipitation experiments also show a significant temperature control on the incorporation of Mg and Sr into calcite (Kinsman and Holland, 1969; Katz, 1973; Oomori et al., 1987). While a positive co-variation of Mg/Ca ratios with ambient
temperature in biogenic calcite agrees with experimental data from inorganic calcite precipitation experiments, Sr/Ca ratios in shell calcite that are positively linked with temperature (e.g., Lea et al., 1999; Stoll et al., 2002; Ullmann et al., 2013a) contradict an expected negative correlation of these two parameters (Kinsman and Holland, 1969; Rimstidt et al., 1998).

It is evident that a multitude of parameters besides temperature can affect element concentrations in biogenic calcite, e.g., pH (Lea et al., 1999), $pCO_2$ (Dissard et al., 2002; Müller et al., 2014), salinity (Klein et al., 1996; Lea et al., 1999; Wanamaker et
al., 2008; Hönisch et al., 2013) and notably calcite secretion rate (e.g., Klein et al., 1996; Lorrain et al., 2005; Ullmann et al., 2015). Disentangling the effect of temperature on shell geochemistry from the effects of physiological responses triggered by temperature change, and constraining the relative contributions of these parameters on shell chemistry, is difficult in growth experiments (Wanamaker et al., 2008). For micro-organisms analyses need to be performed at high spatial resolution and inter-specimen offsets become a concern (e.g., de Nooijer et al., 2014). Most shell-building macro-organisms for which
growth rates can be readily established (e.g, Mouchi et al., 2013; Nedoncelle et al., 2013; Pérez-Huerta et al., 2014) form growth increments that do not show consistent, significant, lateral differences in secretion rate.

Belemnites, Mesozoic predators whose fossil calcite is of major importance for the reconstruction of palaeoenvironmental conditions during the Jurassic and Cretaceous (e.g., Podlaha et al., 1998; McArthur et al., 2000; Ullmann et al., 2014; Sørensen et al., 2015), constitute an exception to this. Their rostra are typically large (a few to tens of centimetres long and a
few millimetres to centimetres in diameter) and are structured by a concentric arrangement of growth bands around the apical line, which itself traces the long axis of the fossil (Sælen, 1989; Ullmann et al., 2015). These growth bands show systematic lateral changes in thickness and can be sampled by milling, permitting high-precision analyses of element concentrations via ICP-OES or ICP-MS. In the rostra it is possible to differentiate between effects of crystal morphology, secretion rate and other physiological or environmental controls (Ullmann et al., 2015). A significant contribution of calcite
secretion rate to Mg/Ca and Sr/Ca patterns in belemnite calcite has been noted (Ullmann et al., 2015). The observed negative correlation of Mg/Ca with secretion rate and positive correlation of Sr/Ca with secretion rate in the calcite of a rostrum of the belemnite *Passaloteuthis bisulcata* agrees with published experimental studies (Lorens, 1981; Tesoriero and Pankow, 1996; Gabitov and Watson, 2006; Tang et al., 2008; Gabitov et al., 2014) and theoretical studies (DePaolo, 2011; Gabitov et al., 2014). The magnitude of secretion rate-induced changes of Mg/Ca and Sr/Ca ratios, however, has so far remained
unquantified in belemnite calcite.

Here, a quantitative appraisal of the available data is presented, backed up by additional qualitative data and the importance of calcite secretion rate for the interpretation of Mg/Ca and Sr/Ca ratios in biogenic calcite is discussed.

## 2 Materials and Methods

The studied, nearly complete rostrum of *Passoleuthis bisulcata* (Blainville 1827) was collected from the Early Toarcian (Early Jurassic) Grey Shale Member at Hawsker Bottoms, Yorkshire, UK (67 cm above the base of the *Dactylioceras tenuicostatum* ammonite subzone, *D. tenuicostatum* zone; Hesselbo and Jenkyns, 1995). Detailed methodology for geochemical analyses and documentation of cathodoluminescence patterns is described in Ullmann et al. (2015). In brief, the specimen was cut perpendicular to the long axis into four slabs which were glued onto glass slides, ground down to a thickness of ca. 3.5 mm and polished. Samples for element/Ca analysis were prepared by incremental drilling of samples (ca. 110 μm increments; ca. 2 mm drill depth; ca. 0.8 mm drill bit diameter) through the slabs using a hand held drill. Analyses were performed using an Optima 7000 DV ICP-OES (Fig. 1). After grinding down the sampled slabs to ca. 0.5 mm and polishing, cathodoluminescence maps were made using a microscope equipped with a Citl Mk-3a electron source (Fig. 2). Element/Ca ratios were screened for diagenesis using Mn/Ca ratios and only data from well-preserved samples are further considered.

Data from the specimen of *Passaloteuthis bisulcata* analysed at high resolution are supplemented by geochemical results of an additional transect through another specimen of *Passaloteuthis* sp. from the Grey Shale Member at Hawsker Bottoms (1 cm above the base of the *Dactylioceras tenuicostatum* ammonite subzone, *D. tenuicostatum* zone; Hesselbo and Jenkyns, 1995; Table S1). A section of the specimen close to the phragmocone (comparable to profile one in Fig. 1) was prepared as described above and sampled along a traverse with 300 μm spacing and drill depth of 500 μm using a MicroMill with 0.6 mm diameter drill bit. Resulting powders were dissolved in weak HCl and diluted to a nominal Ca concentration of 10 μg/g with 2 % $HNO_3$ and analysed for Mg/Ca, Sr/Ca and Mn/Ca ratios using a Perkin Elmer Elan Quadrupole ICP-MS at the University of Oxford. Quantification of concentrations was performed using a set of matrix-matched, synthetic calibration solutions mixed from single element solutions. Accuracy and precision were assessed by multiple analyses of the international reference material JLs-1 and internal carbonate standards, and long-term reproducibility over a period of 3 years is found to be ca. 6 % (2 sd, n = 14) for all element ratios. Assessment of sample preservation was performed analogously to Ullmann et al. (2015).

## 3 Results

Cathodoluminescence microscopy reveals a multitude of characteristic, luminescent bands in the rostrum that constitute time stamps which can be correlated through the entire rostrum (Fig. 2). Differences in the distance of these luminescent bands from the apical line in the four profiles can be used to trace the relative secretion rate changes in belemnite rostra (Fig. 2). Due to the position of the different profiles within the rostrum (Fig. 1A), relative secretion rate becomes greater from profile one to four and the central growth bands in profile one are progressively lost (Figs. 1A,B, 2). Profile four yields only the outer ~30 % of the growth bands present in profile one but these growth bands have thicknesses (relative secretion rates) 2.3 to > 3 times greater than the correlative bands in profile one (Fig. 1C). Using exponential functions to express the differences

in secretion rate (Fig. 1C, Ullmann et al., 2015), Mg/Ca and Sr/Ca ratios of the four geochemical profiles can be integrated in a common ontogenetic profile (Fig. 3). These overlays show common patterns but significant offsets between the profiles. All profiles are characterised by a steady decrease of Mg/Ca ratios and Sr/Ca ratios from the innermost growth bands reaching a minimum at ~ 60 % distance from the centre of the rostrum in the reference profile one (Fig. 3). Both ratios then increase until about 75 % distance on the reference profile and towards the rim show a minor decrease with subordinate peaks and lows which are specific to Mg and Sr. In addition to this general pattern, progressively lower Mg/Ca and higher Sr/Ca ratios are observed at the margin of the rostrum with increasing profile number. Differences in element/Ca ratios with respect to the reference profile one as a function of changes in calcite secretion rate are plotted in Figure 4. For these plots, only geochemical data from the outer 2.0 mm (profile two) to 2.6 mm (profile four) were taken into account, where differences between the profiles are thought to be dominated by secretion rate effects (Ullmann et al., 2015). Deviations in Mg/Ca and Sr/Ca from the reference profile co-vary strongly and the resulting enrichment factors show strong co-variation with relative secretion rate (Figs. 4A–C). Best fits for these relations are:

$$\Delta \frac{Mg}{Ca} = (0.994 \pm 0.007) - (0.081 \pm 0.009) * \Delta\ secretion\ rate \tag{1}$$

$$\Delta \frac{Sr}{Ca} = (1.006 \pm 0.006) + (0.059 \pm 0.007) * \Delta\ secretion\ rate \tag{2}$$

$$\Delta \frac{Mg}{Ca} = (2.37 \pm 0.04) - (1.37 \pm 0.04) * \Delta \frac{Sr}{Ca} \tag{3}$$

where $\Delta$(Mg/Ca) and $\Delta$(Sr/Ca) are the enrichment factors for the element/Ca ratios and $\Delta_{secretion\ rate}$ is the deviation in secretion rate from profile one (0 = 0 %; 1 = 100 %). A secretion rate increase of 100 % thus results in a (8.1 ± 0.9) % depletion in Mg and a (5.9 ± 0.7) % enrichment in Sr.

Geochemical data for the transect through an additional specimen of *Passaloteuthis* sp. (Figs. 5,6, Table S1) confirm the trends observed here and by Ullmann et al. (2015) qualitatively: Diagenesis traced by enrichments in Mn is confined to the apical zone and the dorsal margin and absolute values of Mg/Ca and Sr/Ca ratios vary independent of each other. Enrichments of Mg and Sr towards the apical zone as well as a combination of higher Sr/Ca and lower Mg/Ca ratios in the faster growing, thicker dorsal part as compared to the ventral part of the transect are observed (Fig. 6).

## 4 Discussion

### 4.1 Shell secretion of belemnite rostra and its utility to test growth rate effects

In order to function as a biological experiment tracing the effects of shell secretion rate on Mg and Sr concentrations, the calcite must have been formed incrementally by the belemnite and the secretion rate signal must be large enough not to be masked by other controls. Many aspects of belemnite biomineralisation remain obscure and there is still some ongoing

debate about the original mineralogy (aragonite or calcite) and biomineral architecture of belemnite rostra (e.g., Dauphin et al., 2007; Hoffmann et al., 2016; Immenhauser et al., 2016).

### 4.1.1 Original shell mineralogy and possible porosity

An originally aragonitic rostrum of *Passaloteuthis* analogous to suggestions of Dauphin et al. (2007) for Late Cretaceous *Goniocamax* can confidently be excluded, because the originally aragonitic phragmocones of *Passaloteuthis* are often found to be replaced by minerals other than calcite at Hawsker Bottoms (e.g., pyrite or barite), whereas these phases have never been observed to replace growth bands in the rostra. Also a 50-90 % original porosity of the bulk of the rostrum which was cemented without direct control of the belemnite as envisioned by Hoffmann et al. (2016) is unlikely for *Passaloteuthis*. A restricted zone of original porosity in the apical zone of this genus (Fig. 1A) has been proposed (Ullmann et al., 2015), because here geochemical data fall on a mixing trend with early diagenetic cements of the phragmocone, a signal that is not observed elsewhere in the rostrum apart from its contact with the surrounding sediment matrix. *Passaloteuthis* rostra from Hawsker Bottoms still preserve their original intra-crystalline organic matrix as evidenced by their translucent brown colour and pleochroic behaviour when observed under polarized light (Ullmann et al., 2014). Growth increments can be traced at a resolution of < 10 μm throughout the entire rostrum using cathodoluminescence microscopy. The relative positions of the luminescent bands faithfully define the outline of the rostrum at a given ontogenetic stage and their luminescence intensities are the same in the four profiles without any indication of the typical bright luminescence indicative of early diagenetic cements at Hawsker Bottoms (Ullmann et al., 2015, Fig. 2). Isotope and element patterns can be consistently correlated using only growth band positions (Ullmann et al., 2015), which should not be possible if the larger part of the calcite was formed without the clear temporal transgression defined by the growth bands. $\delta^{13}C$ values in the studied profiles reach a maximum of +3.3 ‰ V-PDB and $\delta^{18}O$ values a minimum of -2.2 ‰ V-PDB (Ullmann et al., 2015), in good agreement with other shelly fossils in the area (Korte and Hesselbo, 2011). If these values were to represent a mixture of less than 50 % original signal with a greater part of calcite representative of bottom water conditions (lighter $\delta^{13}C$ values and $\delta^{18}O$ of -0.4 ‰ or heavier, Ullmann et al., 2014, 2015), recalculated values for the original calcite would become incompatible with other coeval calcite archives. The sum of these observations suggests that – apart from a small zone around the apical line – the rostrum of *Passaloteuthis* was formed in growth increments of calcite with very little original porosity.

### 4.1.2 Utility of *Passaloteuthis* to test growth rate effects on Mg and Sr

Biological controls on element incorporation into shell carbonate are strong and lead to partly significant intra-species differences (e.g., Gillikin et al., 2005; Ullmann et al., 2013a, 2015; Sørensen et al., 2015; Fig. 7). This biological regulation of element partition coefficients (e.g., Gillikin et al., 2005) is also evident in belemnites, which have high Sr concentrations when compared to other coeval calcite fossils (see discussion in Ullmann et al., 2013b). This problem makes constraining the controls on absolute levels of element/Ca ratios in shell carbonate challenging, but is cancelled out, when comparing coeval growth increments within a single fossil. While metabolic controls lead to systematic ontogenetic changes in Mg/Ca and

Sr/Ca in *Passaloteuthis* (Figs. 3,6), at a given ontogenetic stage these metabolic controls are expressed in the same way at the sites of mineralization at each of the studied profiles. The only anticipated difference is the rate of shell secretion, which can thus be isolated as a geochemical forcing and is systematically faster the closer to the rostrum's apex the profile is laid (Fig. 1).

## 4.2 Controls on element uptake and expected growth rate effects

Calcite precipitation experiments have established that Sr concentration should increase and Mg concentrations decrease with increasing precipitation rate (Lorens, 1981; Tesoriero and Pankow, 1996; Gabitov and Watson, 2006; Tang et al., 2008; Gabitov et al., 2014). The same signature is expected to be imposed by decreasing temperature (Rimstidt et al., 1998), necessitating the measurement of a reliable temperature proxy alongside the element concentrations or comparing coeval growth increments. The latter approach is adopted here, which rules out that temperature can have a significant effect on the observed trends. Metabolic effects on element incorporation and/or changes in the composition of the mineralising fluid throughout ontogeny are clearly evident (Figs. 3,6), but are not manifested as a strong co-variation of Mg and Sr. These effects can thus be accounted for by normalising element/Ca data to element/Ca ratios of profile one, i.e., by computing element enrichment factors. Furthermore, changes in relative growth rate (within a factor of four) and likely also absolute growth rate in the rostrum are not very large, so that the observed effects are comparable throughout the studied part of the profiles. They therefore image a linear segment of a relationship, which over a wider range of secretion rates is expected to follow a more complex, curved function (e.g., Tang et al.; DePaolo, 2011; Gabitov et al., 2014).

The observed sensitivities of Sr/Ca ratios and Mg/Ca ratios to changing secretion rate (5.9 % increase and 8.1 % decrease per 100 % shell secretion rate increase) can be compared with experimental results supported by theoretical considerations. The growth entrapment model (Watson and Liang, 1995) predicts that elements present in the surface layer of a growing crystal become more efficiently trapped the faster the crystal forms. Results of precipitation experiments at 20–25°C approximated with this model Gabitov et al. (2014) predict a Sr/Ca sensitivity observed in *P. bisulcata* for calcite growth rates of ca. 0.05 nm s$^{-1}$ and ca. 40 nm s$^{-1}$, whereas measured Mg/Ca sensitivity is matched at ca. 0.3 nm s$^{-1}$ and ca. 20 nm s$^{-1}$. It is conceivable that at slightly different temperatures a better match between these two elements could be obtained, because the response of Sr changes significantly with temperature (Tang et al., 2008). No equivalent experiments for Mg are available, however, so this hypothesis cannot be quantitatively explored. Nevertheless, conditions can be found in experimentally constrained relationships under which the proposed sensitivities of Mg and Sr to shell secretion rate change recorded in *P. bisulcata* are met, further supporting that relative secretion rate is the ultimate control of the observed signal in Fig. 4.

## 4.3 Significance of shell secretion rate effects for fossil Mg/Ca and Sr/Ca data

In order to use Mg/Ca and Sr/Ca data of fossil carbonates to study aspects of palaeoenvironments, it is imperative that the dominant controls of the signal are constrained. After excluding data that are affected by crystallographic forcing near the centre of the rostrum (Ullmann et al., 2015, Fig. 6) and samples showing clear secretion rate effects, the residual range

between the measured extreme values in the studied rostrum is still $\pm$ 25 % for Mg/Ca ratios and $\pm$ 12 % for Sr/Ca ratios (Fig. 7). Reported variability within the genus *Passaloteuthis* and all Toarcian (Early Jurassic, ca. 183–174 Ma) belemnite rostra (Bailey et al., 2003; Rosales et al., 2004; Ullmann et al., 2014) is considerably larger (Fig. 7). Some of this variability is likely related to crystallographic controls on Mg and Sr incorporation (Fig. 7, Ullmann et al., 2015). In practice, there are

only a few samples affected by this factor in published data sets: Crystallographic controls leading to such Sr and Mg enrichments are most prevalent near the apical line of the rostrum (Ullmann et al., 2015), an area where sampling is avoided (if possible) because of a high probability of diagenetic overprint (e.g., Podlaha et al., 1998; McArthur et al., 2000; Ullmann and Korte, 2015). For *Passaloteuthis*, the observed range of Mg/Ca and Sr/Ca values in the genus (Ullmann et al., 2014) is reduced by a third, when the highest 5 % of the element/Ca ratios are excluded.

Significant changes of average Sr/Ca and Mg/Ca ratios have been observed for whole belemnites specimens throughout the Toarcian (McArthur et al., 2000; Bailey et al., 2003; Rosales et al., 2004). These changes are considerably larger than observed intra-specimen variability (Fig. 7) and show a large increase of both Sr/Ca and Mg/Ca ratios around the Early Jurassic Toarcian Oceanic Anoxic Event (ca. 183 Ma, McArthur et al., 2000; Bailey et al., 2003; Rosales et al., 2004). On an even longer time scale of the Early and Middle Jurassic – some 38 million years – Sr/Ca ratios in belemnite rostra averaged

per ammonite biozone drift between 1.2 and 2.3 mmol mol$^{-1}$, probably tracing secular changes in seawater composition (Ullmann et al., 2013b).

While effects of secretion rate on Sr/Ca and Mg/Ca ratios in belemnite rostra are significant (Fig. 4,6), all the above described ranges of data are considerably larger than the maximum observed effect of calcite secretion rate in *P. bisulcata* (Fig. 7), suggesting that growth rate is of minor importance for controlling element/Ca ratios in this species. This likely also

holds for other belemnites as seen by Sr/Ca ratios in transects through multiple specimens yielding comparable results in *Bellemnellocamax mammillatus* (Sørensen et al., 2015), even though the generalisation remains to be tested rigorously. When sampling belemnite rostra for palaeoenvironmental studies, avoiding the apical zone and targeting profiles as far away from the apex as possible (i.e., profile one rather than profiles three or four) ensures the least possible bias exerted by crystallographic forcing and internal growth rate effects. Deriving information about past seawater composition then depends

mostly on other potential controls on shell chemistry like metabolic, kinetic, and temperature effects (e.g. Rosales et al., 2004; McArthur et al., 2007; Li et al., 2012) and the generation of large datasets which enable constraining average values precisely.

## 5 Conclusions

The biomineral structure of belemnite rostra provides a unique opportunity to isolate effects of calcite secretion rate from

other controlling processes during biomineralisation.

Changes in relative growth rate in the rostrum of the belemnite *P. bisulcata* result in a depletion of Mg (8.1 $\pm$ 0.9 % per 100 % secretion rate increase) and enrichment of Sr (5.9 $\pm$ 0.7 % per 100 % secretion rate increase).

While the forcing exerted by changing secretion rate is significant and can be precisely quantified, it is of minor importance for the overall variability of Mg/Ca and Sr/Ca ratios in belemnite rostra. This is important when considering the utility of belemnite rostra for palaeoenvironmental analysis.

Sampling geochemical transects along profiles as near as possible to the protoconch of the rostrum and avoiding the apical

zone helps minimising biasing effects of secretion rate (and crystallographic forcing) on palaeoenvironmental datasets.

**Acknowledgements**

Analyses and PPvS were funded by NERC research fellowship grant NE/I020571/2. CVU acknowledges funding from the Leopoldina – German National Academy of Sciences (grant no. LPDS 2014-08). Kate Littler is thanked for comments on an earlier version of this manuscript. The authors thank the Associate Editor Dr. David Gillikin, Prof. Adrian Immenhauser and

one anonymous reviewer for constructive comments that helped to significantly improve the quality of the manuscript.

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

**Figure 1: Studied geochemical profiles and their relative secretion rates. A) Schematic representation of the rostrum of *P. bisulcata* with approximate positions of the transects. B) Cross plot of distance of selected luminescent bands (a–f and white dots from Fig. 2) from the margin of the rostrum in profiles one to four with respect to its position in profile one. Stippled lines indicate selected relative secretion rates with respect to profile one. Letters and circles are positioned as in Figure 1. C) Relative secretion rate in geochemical transects computed from exponential relationships documented in Ullmann et al. (2015).**

**Figure 2: Cathodoluminescence pattern for sections through the rostrum adjacent to geochemical transects one to four. Letters "a" to "f" correlated between the profiles with white lines as well as white dots indicate arbitrarily chosen marker bands for illustration of differences in shell secretion rate (see also Fig. 2). Note the increasing distance of marker bands from profile one to four.**

**Figure 3: Aggregated geochemical data for the four profiles plotted against distance from the central apical line of the rostrum (zero) to the margin (one) with profile one as a reference. The bin size is 2.5 % of the profile length. A) Mg/Ca ratios. B) Sr/Ca ratios.**


**Figure 4: Changes in Mg/Ca and Sr/Ca ratios as a function of precipitation rate. A) Mg enrichment factor as a function of change in precipitation rate expressed as a relative deviation from the reference precipitation rate of profile 1. Vertical lines denote 2 standard error uncertainties for each binned interval of profiles two to four and circles show average values with two standard error uncertainty. The trend line from ordinary least square regression of the mean values is shown with 95 % uncertainty envelope. B) Sr enrichment factor as a function of calcite precipitation rate. Symbols as in A). C) Correlation of Mg depletion with Sr enrichment.**


**Figure 5: Section through additional specimen of *Passaloteuthis* sp. with sample positions for geochemical analyses.**

**Figure 6: Geochemical data for additional specimen of *Passaloteuthis* sp. (Mg/Ca green; Sr/Ca violet; Mn/Ca brown). Samples excluded from interpretation due to postdepositional alteration (Ullmann et al., 2015) are shaded in gray. Significant differences in average Sr/Ca and Mg/Ca between the ventral (slow growing) and dorsal (fast growing) part of the section (light straight lines) are evident and are compatible with the findings from the multi profile dataset.**

**Figure 7: Variability of Mg/Ca and Sr/Ca ratios in belemnite calcite. Intra-specimen variability accounts for residual range of values in *P. bisulcata* after accounting for crystallographic forcing and shell secretion rate (Ullmann et al., 2015). Intra-species variability of *Passaloteuthis* (Ullmann et al., 2014) and range of values observed in Toarcian belemnites (Bailey et al., 2003; Rosales et al., 2004; Ullmann et al., 2014) do not exclude samples potentially affected by crystallographic forcing and are thus likely overestimated. Range of reconstructed seawater Sr/Ca ratios is from Ullmann et al. (2013b). Length of arrows for crystallographic forcing and secretion rate indicate maximum observed effect in *P. bisulcata*.**

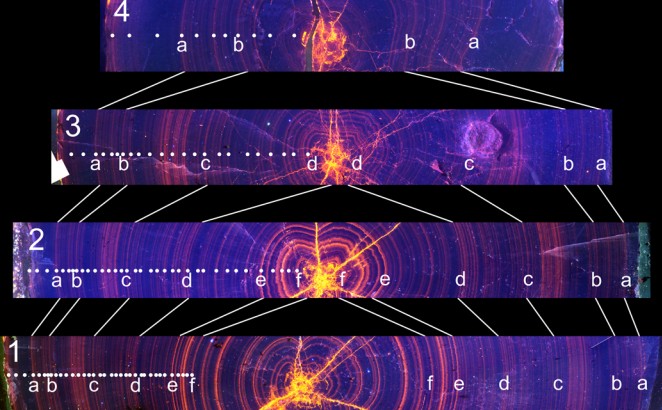

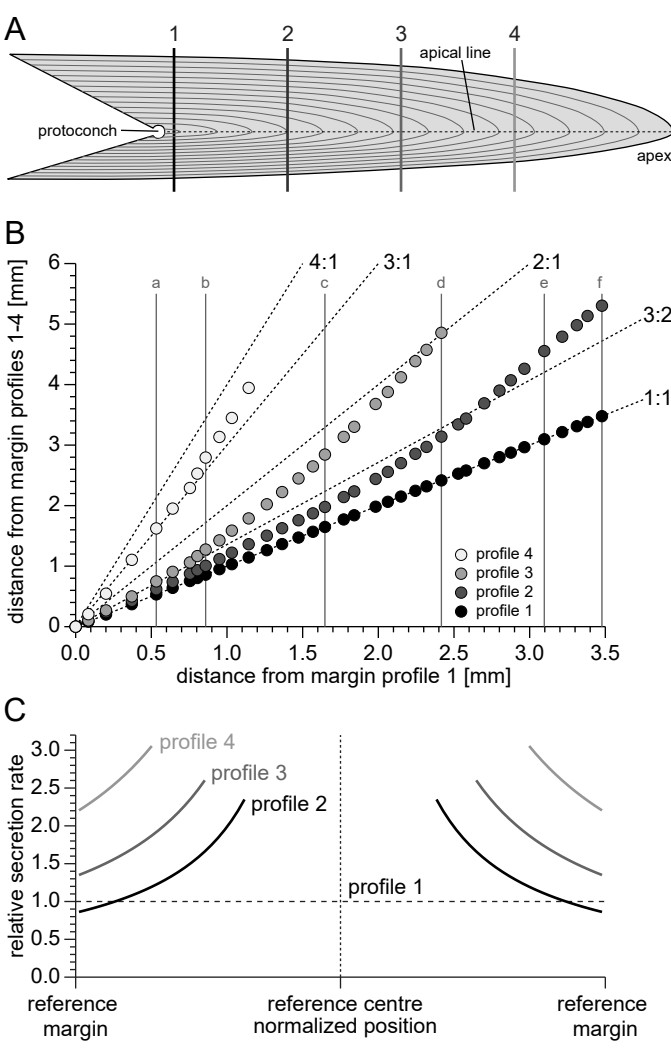

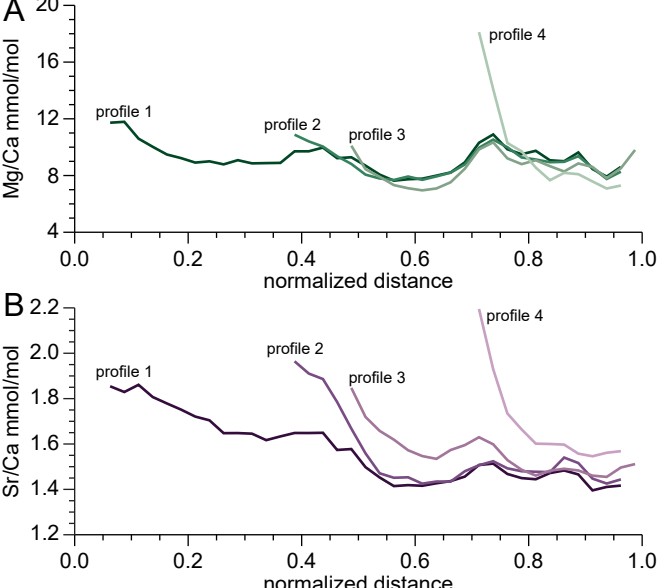

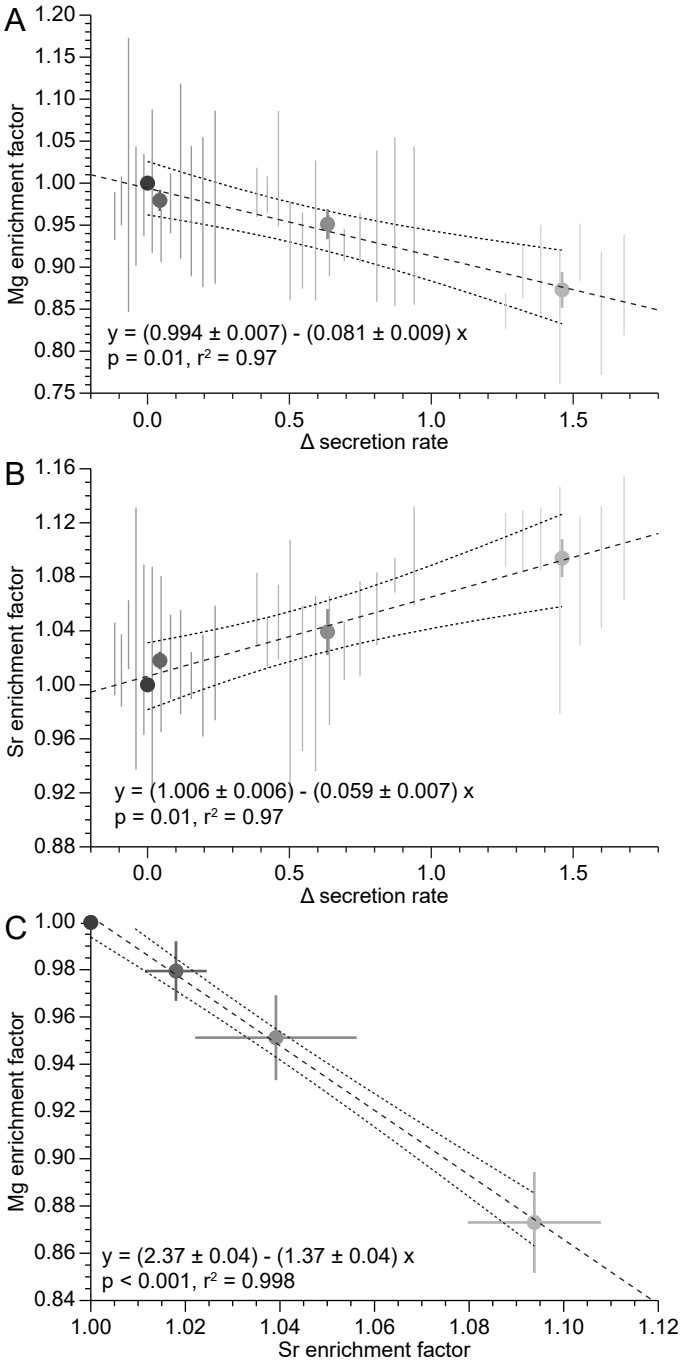

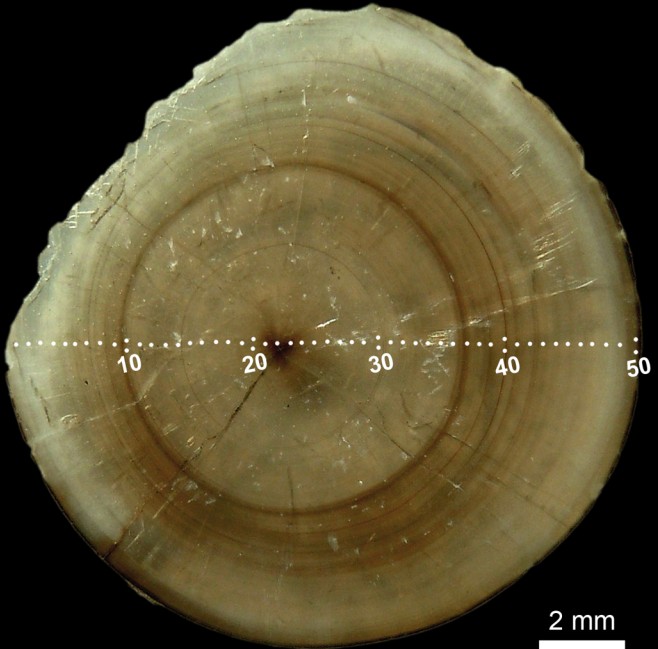

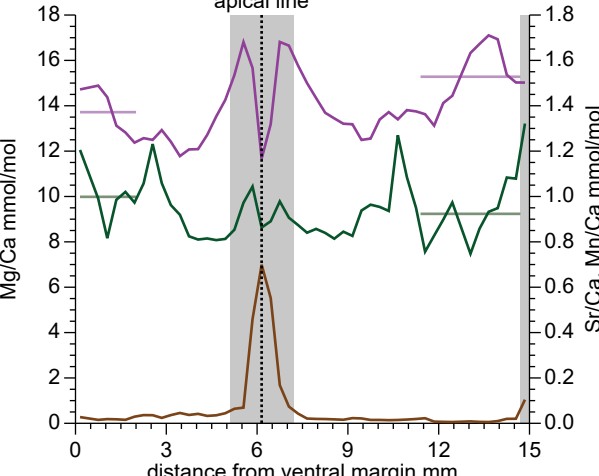

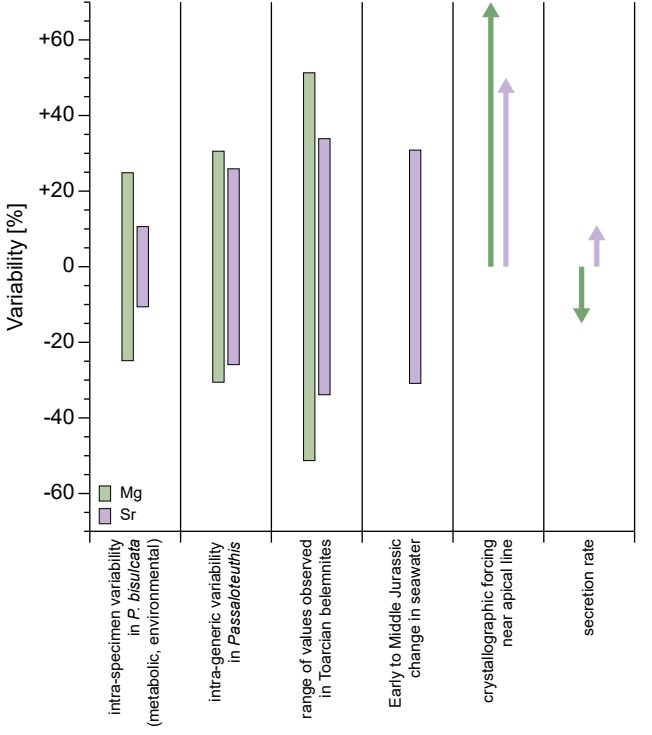