# Peer review of "The effect of shell secretion rate on Mg/Ca and Sr/Ca ratios in biogenic calcite as observed in a belemnite rostrum"

_Biogeosciences, 2016_

## Referee Comment (RC1) · A. Immenhauser (Referee) · 19 Sep 2016

Dear author, dear editor,

Thank you for providing me with the opportunity to review this paper. Dr. Ullmann reports on his statistical attempt to capture calcite secretion rates in belemnite rostra and specifically on the kinetic effects of secretion rates on Mg/Ca and Sr/Ca ratios. Dr. Ullmann is a recognized expert in this research field and has given this issue significant consideration. The main outcome of this study is that the effects of kinetics are comparably minor and can be avoided by an intelligent sampling strategy. In essence, I feel that this paper has significant merits and should be published pending what I consider significant revisions. My comments are simply suggestions to make this an even better

paper.

Below I summarize my main concerns:

I used the search option of Adobe Acrobat pro but did not find the word "biomineralization" mentioned a single time in this paper (I find metabolism once). The same accounts for "kinetics". I find this surprising. Similarly, the biomineralization by the belemnite animal is referred to as "precipitation". In my view, the manner in which organisms form their endo- or exoskeletons is referred to as "carbonate secretion". I can live with all of that as this is essentially terminology. Nevertheless, please consider.

Similar to many other metazoan biomineralizers, molluscs isolate their environment of mineral formation from the outside world. Page 2 refers to known parameters that affect Mg/Ca ratios in inorganic precipitation experiments but these are apparently placed on an identical level as those that govern the body fluids of the animal? I am very critical here!

I agree with the statement in the abstract: It is often hard to constrain which parameter ultimately controls the concentrations of a given element in biogenic calcite. Ambient environmental parameters affect the physiology and metabolism of the animal and hence there is a strong correlation between environment and metabolisms. What controls carbonate secretion rates of the belemnite rostra: Environmental parameters, food availability, ontogenetic trends, stressors and more of the like.

Moreover, there are the issues of possible stressors, sexual dimorphism and species-specific biological controls on carbonate secretion rates. In my view, but I might be wrong, the problem is that many parameters control the rates of carbonate secretion, which in turns affects – via kinetics – element incorporation. But in parallel to this, element incorporation in turn is affected by factors other than kinetics too. I understand that Dr. Ullmann is approaching this by means of least square regression approaches that are helpful but will not solve all problems. This is very briefly touched upon on page 5 and the author concludes that the observed fluctuations are such that they exceed

intra-specimen variability and hence represent secular "global" environmental patterns. I am not sure I am an advocate of the concept that the world's oceans in the geological past did see a uniform change in their parameters? A look on the data sets provided by oceanographers reveals a very complex chemical and physical structure of the present day oceans. Plenty of heterogeneity and regional trends!

I am not sure if the author should use the label "quantitative appraisal" here? I am not convinced he can clearly separate the effects of kinetics from the bulk of parameters that govern element incorporation in these biominerals. I would agree with Dr. Ullmann, that his approach represents the perhaps best amongst all of the less-than-ideal approaches by which this difficult issue can be approached. Using sophisticated statistical tools on a non-sophisticated data set helps, that is clear, but this does not implies that the result is quantitative. Moreover, given that appraisal is often used as synonym for assessment or even opinion, the term seems somewhat contradictory.

Summing up: I applaud Dr. Ullmann in his attempt to get a better grip on the effects of kinetics on elemental ratios in belemnite ratios. In the past, many of us have assigned these proxy data to temperature alone in a rather uncritical manner, or when data were difficult to interpret, to either "biological" effects or "diagenesis". The latter two representing popular black box interpretations. This paper represents a clear progress and the essential message brought forward is important and valid and this is why the paper should be published. I believe the paper suffers from a – in my view – uncritical application of results from inorganic precipitation experiments to biomineralization in the body of an organism. I believe the paper suffers from a selective choice of references. Yes, it is true, a number of empirical studies have documented co-variation of Mg/Ca and Sr/Ca ratios in these carbonates with what was assumed to be temperature (or in the case of recent organisms) has been measured as ambient temperature. It holds also true, however, that important papers have shown aquaria or field experiments that reveal the full complexity of these biogenic archives and kinetics was but one. What would this imply? I find these critical voices underrepresented in the present paper.

That surprises me given that the author has chosen to investigate one of the many other-than-temperature effects, here kinetics, as a topic of his paper. It seems "two souls are dwelling in his chest"? So, the messages are: (i) We can ignore kinetics and (ii) belemnites are still our favourite archive organisms for the Mesozoic?

Finally, it is certainly not my style to use a review to make an author cite my papers! Nevertheless, our recent paper in Sedimentology (Immenhauser et al. 2016) provides a wide selection of references that can be cited here and problems that should be considered. I believe that Dr. Ullmann knows this paper? Please also consider making reference to Benito et al. (2016; J. Iberian Geology, 42, 201-226 and Hoffmann et al. (2016, Sed Geol, 341, 203-215). It is perfectly fine, if you disagree with these authors, but ignoring them totally might reflect poorly on this paper.

I hope these comments are of use!

Sincerely yours,

A. Immenhauser

---

## Referee Comment (RC2) · Anonymous Referee #2 · 12 Oct 2016

REVIEW: The effect of precipitation rate on Mg/Ca and Sr/Ca ratios in biogenic calcite as observed in a belemnite rostrum by Clemens Vinzenz Ullmann

Given the importance Mg/Ca and Sr/Ca ratios in biogenic carbonate as paleoenvironmental indicators, Ullmann investigates how skeletal precipitation rates influence incorporation of Mg and Sr in a belemnite rostrum. Ullmann observes that as precipitation rate increased Mg/Ca ratios decreased and Sr/Ca ratios increased. Additionally, Ullmann observed that elemental ratios covaried linearly with precipitation rate. However, since this relationship does not account for all of the geochemical variation seen in the skeleton, with careful sampling, obtaining reliable geochemical data that reflects past environmental conditions may be possible. This paper presents important findings on the importance of understanding how biology influences paleoenvironmental proxies archived in biogenic carbonate. The marerials and methods are, for the most part, sound (but see below). The manuscript is clear and the writing is excellent. The references are appropriate and the figure are excellent. The most significant problem with this paper is that the findings are based on a single specimen. While that data and interpretations are consistent with the patterns in this belemnite. Without corroborating the pattern with data from additional individual, then significance of this study is limited. I recommend that Ullman replicate his findings with additional specimen(s) before publishing this study.

---

## Author Comment (AC2) · 15 Oct 2016

Many thanks for this review.

Referee 2 critically points out that this study relates to the signals of only one fossil and that these findings are therefore of limited extent.

It is true that the data for this study – even though > 300 ICP-OES analyses were conducted – relate only to a single rostrum. Despite the self-consistent results verified by three profiles compared to a reference profile and tentative support from geochemical profiles measured for other belemnite species it is at present not self-evident that a generalization of these patterns is possible. Given the consistency of the dataset and

independent support from other lines of carbonate research I am confident, however, that this pattern holds.

I intended this study to be a proof-of-concept, to introduce a biomineralization system which deserves further research and to point out where I see synergies between research on abiogenic and various biogenic calcite archives. I would be delighted to see my initial results reproduced in other belemnite species and – if possible – tested in other biogenic calcite archives as well.

There is one particularly thrilling outcome I see in the presented dataset: If the temperature dependence of the precipitation rate forcing on Mg, Sr and other elements can be constrained more precisely, my approach yields a novel palaeotemperature proxy (section 4.1). This temperature proxy would be independent of seawater chemistry and much more robust against diagenesis and easier to measure than clumped isotopes: Once it is known what the rate forcings on elements like Mg, Sr and Li during coprecipitation with calcite are at various temperatures, one can potentially establish an overconstrained system for the two unknowns, 1) absolute precipitation rate and 2) temperature.

---

## Author Response (AR1)

Associate Editor (Dr. David Gillikin):

1) You cannot draw these conclusions based on one specimen. There are several published papers showing large differences in mollusk skeleton elemental ratios between specimens and species. Vital effects can result in large differences in relationships between elements and precipitation rates and even between different elements even within the same species. For example, look at the differences in Mg/Ca ratios between scallop specimens in Lorrain et al. 2005 cited in your manuscript. Also, consider the large differences in Sr vs precipitation rate between species shown in my own paper, Gillikin et al. 2005 ( doi:10.1029/2004GC000874). I urge you to include more data from a second specimen at a minimum. You need to replicate these patterns to draw the conclusions you discuss. With additional data your manuscript will be more well accepted by the community.

In the revised manuscript, data for an additional specimen of Passaloteuthis (end of Sections 2, Materials and Methods and 3, Results) are presented which show the same trends as the specimen for which multiple high resolution profiles were studied. These data were generated in collaboration with Dr. Philip A.E. Pogge von Strandmann who is now included as a co-author on this manuscript.

Additionally, a new paragraph (section 4.1.2) was added to the discussion chapter, where the unique angle of the analysis in this manuscript is laid out in more detail: It Is acknowledged that considerable inter-specimen differences exist with respect to element composition and response to growth rate change in carbonate shells. This is also evident for the genus Passaloteuthis (Fig. 7 in the manuscript), even though average Sr/Ca and Mg/Ca of the two specimens reported here are nearly the same. Such differential metabolic response and changes in growth rate resulting from external forcings are of little concern to the present study, however. Here, the focus is on the comparison of shell secretion within in a single organism at different shell formation rates at discrete time intervals where differential effects of external forcings and metabolic controls (apart from shell secretion rate) are cancelled out.

Reviewer 1 (Prof. Adrian Immenhauser):

1) Adoption of termini "biomineralization"; "kinetics"; "carbonate secretion"
The terminology in the manuscript has been revised according to these terms and "precipitation rate" has been reformulated unless when abiogenic experiments are referred to.

2) Comparability of inorganic with organic systems (experimental fluid vs body fluid)
I agree that processes of shell formation and composition of body fluids are much more complex than what can currently be modeled using inorganic precipitation experiments. It was not my intention to advocate that inorganic experiments and growth experiments of molluscs are entirely interchangable or that the former can replace the latter.

However, I feel that one can count it as an encouraging sign if empirical results on biological systems and experimental results on abiogenic precipitates converge. After all, most of palaeoenvironmental geochemistry relies on the assumption that fundamental physical controls on carbonate composition hold - regardless which animal (or "non-animal") produced it. Quite a substantial body of literature employing this underlying assumption and yielding results consistent with non-geochemical interpretations of the rock record has emerged so that one can be confident that some truth is in this assumption.

I am of the opinion that one should incorporate as broad as possible a range of studies when investigating biomineralization signals whilst keeping their limited comparability in mind.

3) Multiple controls and feedbacks of shell secretion and kinetics
Prof. Immenhauser rightly points out that a multitude of environmental parameters, food availability, ontogenetic trends, stressors, sexual dimorphism, species specific effects, and many more have an effect on the chemical composition of biogenic shell materials. Many of these parameters will also have a bearing on shell secretion rate which in turn affects element incorporation, making biomineralization a very complex phenomenon.

I entirely agree with Prof. Immenhauser in this point, but what I investigated in the present study is a system, in which *relative* shell secretion rate can be viewed upon as an isolated parameter. I did not claim that this study is able to make inferences about *absolute* shell secretion rates or to make a point about the environmental and biological factors that control shell secretion rate.

The unique angle of the present study is to look at time slices of belemnite shell secretion. The assumption is that a growth band in a belemnite represents an ontogenetic isochron for which one particular expression of biomineralization controls is realized. The growth band, which then by definition has to represent the same amount of time, has a variable thickness along the whole of the belemnite rostrum, i.e., the rate at which it formed is variable for purely geometrical reasons, increasing from the alveolar area to the apex (Figs. 1, 2). By normalizing the chemical data of all measured profiles against data from a reference profile, all external controls on shell chemistry are then taken out of the equation and it becomes possible to isolate what the effect of this rate variability is.

This is now elaborated on in Discussion section 4.1.2.

4) Least squares regression doesn't solve all problems. Geochemical variability of fossil geochemistry may not represent secular global trends - heterogeneity and regional trends in oceanographic parameters have to be considered.
It is true that it is only an inference that the chemical composition of fossil materials can be

used to reconstruct the chemical composition of seawater through time.

I am confident that this is possible for some elements to a certain degree to reconstruct past seawater composition. The elements Mg, Sr and Ca which are looked at in this study are amongst those which are thought to be uniformly distributed in the oceans due to their long residence time. Their concentrations and even their isotopic ratios vary within very narrow limits in the world oceans and this is unlikely to have been different in the past due to the particular chemical behaviour of these elements. For the above elements one can therefore neglect regional trends in fully marine environments – one reason e.g. for the success of the marine Sr isotope curve. For elements with much shorter oceanic residence times like Ce, Cr, Cd, Mn etc. it would be an entirely different story.

One observation which appears to be temporally robust is that different Sr/Ca ratios in marine shells have certain relative offsets from the seawater Sr/Ca ratio they form in. Even though it is very seldom utilized, a crude chemical mapping of the composition of modern biominerals is available since 50 years (Dodd, 1967, JOURNAL OF PALEONTOLOGY). The first order observation is that Sr/Ca of average bivalve calcite is lower than the Sr/Ca ratio of the average brachiopod, which in turn (for Jurassic and Cretaceous) is lower than the Sr/Ca ratio of the average belemnite (Voigt et al, 2003, INT J EARTH SCI; Korte and Hesselbo, 2011; PALEOCEANOGRAPHY; Ullmann et al., 2013, GEOLOGY; Sørensen et al., 2015, PPP; Ullmann et al., 2016, GONDWANA RESEARCH). For brachiopods one finds Sr/Ca ratios decreasing from craniids to thecideids to terebratulids and rhynchonellids (Brand et al., 2003, CHEM GEOL). Within the bivalves, oysters have a lower average Sr/Ca than pectinids. This is subject to ongoing research and could be refined ad infinitum. Once the translation factor from element/Ca(fossil) to element/Ca(seawater) is known and can be reproduced through time with multiple fossil groups, the likelyhood is that the seawater element/Ca ratio thus studied can be confidently reconstructed.

What is imaged in Fig. 5 (now Fig. 7) is the "biomineralization noise", one has to overcome for such an approach. Due to all sorts of intra-specimen, intra-specific, and inter-specific effects a large number of analyses has to be integrated for a meaningful average. Relative shell secretion rate, however, is a small player in this game, and this is the contribution I wanted to make with this study. While one may remain critical of the idea that seawater chemistry reconstructions can be done on the basis of shell chemistry, this study shows that shell secretion rate is not to blame if it does not work, because its forcing is inconsistent with observed data variability and its magnitude is too small.

**5) Is the term "quantitative" justified?**

The short answer is "yes". What is presented is a quantification of the effect of relative shell secretion rate on the Mg/Ca and Sr/Ca ratios found in a belemnite rostrum. I find that calcite that forms twice as fast as calcite at another point in the same growth increment will contain 8.1 +/- 0.9 % less Mg and 5.9 +/- 0.7 % more Sr than its slower-forming counterpart – regardless of the absolute Mg and Sr concentrations which are subject to more complex forcings (Figs. 3, 4). In this sense this appraisal is quantitative and reasonably precise. These quantitative data are now also backed up by qualitative data from another specimen.

What cannot be quantified is which parameters led belemnites to change their shell secretion rates through their lives.

**6) The messages are "We can ignore kinetics (i.e. secretion rate)" and "belemnites are Mesozoic favourites"?**

Message 1) is a first order "yes". Indeed, besides showing the utility of belemnite calcite for constraining calcite formation rate controls on element uptake, this particular parameter is of

minor importance for generating the chemical complexities of biogenic shell calcite.

Message number 2) is up for debate. Personally, I am a "belemnite fan" and find their calcite is superior to other fossil archives for certain applications, in particular, where large amounts of calcite are necessary (non-traditional isotopes of trace elements come to mind). However, I don't want to advocate that we should neglect other fossil groups (and non-fossil records) or be uncritical about belemnite rostra. The strength of future research will lie in merging the strings of evidence coming from all available sources.

7) Selective choice of references: Missing critical voices saying everything is swamped by biology or the archive is not valid.

It was not my intention to be uncritical about the use of belemnite rostra to constrain shell secretion rate effects on element uptake and neither did I intend to cite the literature in a biased fashion.

Part of the above allegation might be down to a misunderstanding about the aims of the paper as well: The dataset I present is of some value for constraining the effect of shell secretion rate on element partitioning into calcite, 1) because coeval calcite formed at various growth rates (factor of ca. 3) is available and 2) the signal of this relative secretion rate difference is significant despite overall ontogenetic changes in element/Ca ratios in the rostrum.

One may consider this present dataset "a lucky find" whereas for most other fossil groups trying to constrain shell secretion rate effects would fail. I am sure that many biological systems wouldn't lend themselves to such an analysis because the shell growth geometries aren't ideal and/or the chemical variability of the system prevents any meaningful analysis. This should, however, not be taken as evidence that biomineralization processes cannot be disentangled, it simply calls for selecting the right animal group for the right question.

In the revised version of the manuscript a new section (4.1.1) has been included, where it is critically assessed of what is thought about the belemnite rostrum as a geochemical archive including the articles recently published by the Bochum group (Hoffmann et al., 2016; Immenhauser et al., 2016).

Reviewer 2:

Referee 2 critically points out that this study relates to the signals of only one fossil and that these findings are therefore of limited extent.

It is true that the data for this study – even though > 300 ICP-OES analyses were conducted – relate only to a single rostrum. Despite the self-consistent results verified by three profiles compared to a reference profile and tentative support from geochemical profiles measured for other belemnite species it is at present not self-evident that a generalization of these patterns is possible. Given the consistency of the dataset and independent support from other lines of carbonate research I am confident, however, that this pattern holds. This confidence is boosted by the new, additional data now presented in the study which qualitatively confirm the previous findings.

This study was intended to be a proof-of-concept, to introduce a biomineralization system which deserves further research and to point out where synergies between research on abiogenic and various biogenic calcite archives may exist. It would be fantastic to see these initial results reproduced in other belemnite species and – if possible – tested in other biogenic calcite archives as well.

There is one particularly thrilling outcome I see in the presented dataset: If the temperature dependence of the precipitation rate forcing on Mg, Sr and other elements can be constrained more precisely, this approach yields a novel palaeotemperature proxy (section 4.1). This temperature proxy would be independent of seawater chemistry and much more robust against diagenesis and easier to measure than clumped isotopes: Once it is known what the rate forcings on elements like Mg, Sr and Li during coprecipitation with calcite are at various temperatures, one can potentially establish an overconstrained system for the two unknowns, 1) absolute precipitation rate and 2) temperature.

Relevant changes:

Terminology has been revised throughout the manuscript.
Description of additional specimen (end of sections 2, 3, new Figs. 5 and 6) has been added.
Discussion has been appended by critical assessment of belemnite growth bands and mineralogy (section 4.1.1) and section reliability of the observations (sections 4.1.2).
Acknowledgements have been appended by funding sources to P. Pogge von Strandmann and funding for additional analyses.
References have been appended.

[revised manuscript text omitted]

bar

this